# LEARNING INVARIANT REPRESENTATIONS OF PLANAR CURVES

**Gautam Pai, Aaron Wetzler & Ron Kimmel**
Department of Computer Science
Technion-Israel Institute of Technology
`{paigautam,twerd,ron}@cs.technion.ac.il`

## ABSTRACT

We propose a metric learning framework for the construction of invariant geometric functions of planar curves for the Euclidean and Similarity group of transformations. We leverage on the representational power of convolutional neural networks to compute these geometric quantities. In comparison with axiomatic constructions, we show that the invariants approximated by the learning architectures have better numerical qualities such as robustness to noise, resiliency to sampling, as well as the ability to adapt to occlusion and partiality. Finally, we develop a novel *multi-scale* representation in a similarity metric learning paradigm.

## 1 INTRODUCTION

The discussion on *invariance* is a strong component of the solutions to many classical problems in numerical differential geometry. A typical example is that of planar shape analysis where one desires to have a local function of the contour which is invariant to rotations, translations and reflections like the Euclidean curvature. This representation can be used to obtain correspondence between the shapes and also to compare and classify them. However, the numerical construction of such functions from discrete sampled data is non-trivial and requires robust numerical techniques for their stable and efficient computation.

Convolutional neural networks have been very successful in recent years in solving problems in image processing, recognition and classification. Efficient architectures have been studied and developed to extract semantic features from images invariant to a certain class or category of transformations. Coupled with efficient optimization routines and more importantly, a large amount of data, a convolutional neural network can be trained to construct invariant representations and semantically significant features of images as well as other types of data such as speech and language. It is widely acknowledged that such networks have superior representational power compared to more principled methods with more handcrafted features such as wavelets, Fourier methods, kernels etc. which are not optimal for more semantic data processing tasks.

In this paper we connect two seemingly different fields: convolutional neural network based metric learning methods and numerical differential geometry. The results we present are the outcome of investigating the question: *"Can metric learning methods be used to construct invariant geometric quantities?"* By training with a Siamese configuration involving only positive and negative examples of Euclidean transformations, we show that the network is able to train for an invariant geometric function of the curve which can be contrasted with a theoretical quantity: Euclidean curvature. An example of each can be seen Figure 1. We compare the learned invariant functions with axiomatic counterparts and provide a discussion on their relationship. Analogous to principled constructions like curvature-scale space methods and integral invariants, we develop a multi-scale representation using a data-dependent learning based approach. We show that network models are able to construct geometric invariants that are numerically more stable and robust than these more principled approaches. We contrast the computational work-flow of a typical numerical geometry pipeline with that of the convolutional neural network model and develop a relationship among them highlighting important geometric ideas.

In Section 2 we begin by giving a brief summary of the theory and history of invariant curve representations. In Section 3 we explain our main contribution of casting the problem into the form which

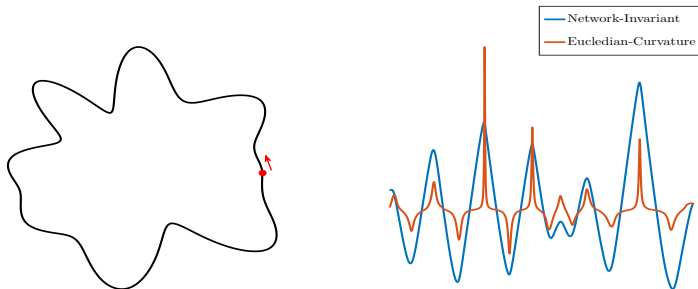

Figure 1: Comparing the axiomatic and learned invariants of a curve.

enables training a convolutional neural network for generating invariant signatures to the Euclidean and Similarity group transformations. Section 4 provides a discussion on developing a multi-scale representation followed by the experiments and discussion in Section 5.

## 2 BACKGROUND

An invariant representation of a curve is the set of signature functions assigned to every point of the curve which does not change despite the action of a certain type of transformation. A powerful theorem from E. Cartan (Cartan (1983)) and Sophus Lie (Ackerman (1976)) characterizes the space of these invariant signatures. It begins with the concept of arc-length which is a generalized notion of the length along a curve. Given a type of transformation, one can construct an intrinsic arc-length that is independent of the parameterization of the curve, and compute the curvature with respect to this arc-length. The fundamental invariants of the curve, known as differential invariants (Bruckstein & Netravali (1995), Calabi et al. (1998)) are the set of functions comprising of the curvature and its successive derivatives with respect to the invariant arc-length. These differential invariants are unique in a sense that two curves are related by the group transformation if and only if their differential invariant signatures are identical. Moreover, every invariant of the curve is a function of these fundamental differential invariants. Consider $C(p) = \begin{bmatrix} x(p) \\ y(p) \end{bmatrix}$: a planar curve with coordinates $x$ and $y$ parameterized by some parameter $p$. The Euclidean arc-length, is given by

$$s(p) = \int_0^p |C_p| \, dp = \int_0^p \sqrt{x_p^2 + y_p^2} \, dp, \tag{1}$$

where $x_p = \frac{dx}{dp}$, and $y_p = \frac{dy}{dp}$ and the principal invariant signature, that is the Euclidean curvature is given by

$$\kappa(p) = \frac{\det(C_p, C_{pp})}{|C_p|^3} = \frac{x_p y_{pp} - y_p x_{pp}}{(x_p^2 + y_p^2)^{\frac{3}{2}}}. \tag{2}$$

Thus, we have the Euclidean differential invariant signatures given by the set $\{\kappa, \ \kappa_s, \ \kappa_{ss} \ ...\}$ for every point on the curve. Cartan's theorem provides an axiomatic construction of invariant signatures and the uniqueness property of the theorem guarantees their theoretical validity. Their importance is highlighted from the fact that *any* invariant is a function of the fundamental differential invariants.

The difficulty with differential invariants is their stable numerical computation. Equations 1 and 2, involve non-linear functions of derivatives of the curve and this poses serious numerical issues for their practical implementation where noise and poor sampling techniques are involved. Apart from methods like Pajdla & Van Gool (1995) and Weiss (1993), numerical considerations motivated the development of multi-scale representations. These methods used alternative constructions of invariant signatures which were robust to noise. More importantly, they allowed a hierarchical representation, in which the strongest and the most global components of variation in the contour of the curve are encoded in signatures of higher scale, and as we go lower, the more localized and rapid changes get injected into the representation. Two principal methods in this category are scale-space methods and integral invariants. In scale-space methods (Mokhtarian & Mackworth (1992); Sapiro & Tannenbaum (1995); Bruckstein et al. (1996)), the curve is subjected to an invariant evolution process where it can be evolved to different levels of abstraction. See Figure 5. The curvature function

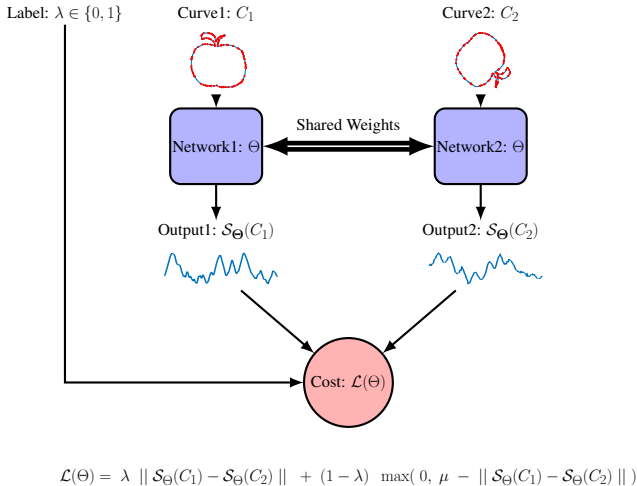

$$\mathcal{L}(\Theta) = \lambda \; \| \mathcal{S}_\Theta(C_1) - \mathcal{S}_\Theta(C_2) \| \; + (1 - \lambda) \; \max(\, 0, \; \mu \; - \; \| \mathcal{S}_\Theta(C_1) - \mathcal{S}_\Theta(C_2) \| \, )$$

Figure 2: Siamese Configuration

at each evolved time $t$ is then recorded as an invariant. For example, $\{\kappa(s,t), \kappa_s(s,t), \kappa_{ss}(s,t)...\}$ would be the Euclidean-invariant representations at scale $t$.

Integral invariants (Manay et al. (2004); Fidler et al. (2008); Pottmann et al. (2009); Hong & Soatto (2015)) are invariant signatures which compute integral measures along the curve. For example, for each point on the contour, the integral area invariant computes the area of the region obtained from the intersection of a ball of radius $r$ placed at that point and the interior of the contour. The integral nature of the computation gives the signature robustness to noise and by adjusting different radii of the ball $r$ one can associate a scale-space of responses for this invariant. Fidler et al. (2008) and Pottmann et al. (2009) provide a detailed treatise on different types of integral invariants and their properties.

It is easy to observe that differential and integral invariants can be thought of as being obtained from non-linear operations of convolution filters. The construction of differential invariants employ filters for which the action is equivalent to numerical differentiation (high pass filtering) whereas integral invariants use filters which act like numerical integrators (low pass filtering) for stabilizing the invariant. This provides a motivation to adopt a learning based approach and we demonstrate that the process of estimating these filters and functions can be outsourced to a learning framework. We use the Siamese configuration for implementing this idea. Such configurations have been used in signature verification (Bromley et al. (1993)), face-verification and recognition(Sun et al. (2014); Taigman et al. (2014); Hu et al. (2014)), metric learning (Chopra et al. (2005)), image descriptors (Carlevaris-Bianco & Eustice (2014)), dimensionality reduction (Hadsell et al. (2006)) and also for generating 3D shape descriptors for correspondence and retrieval (Masci et al. (2015); Xie et al. (2015)). In these papers, the goal was to learn the descriptor and hence the similarity metric from data using notions of only positive and negative examples. We use the same framework for estimation of geometric invariants. However, in contrast to these methods, we contribute an analysis of the output descriptor and provide a geometric context to the learning process. The contrastive loss function driving the training ensures that the network chooses filters which push and pull different features of the curve into the invariant by balancing a mix of robustness and fidelity.

## 3 TRAINING FOR INVARIANCE

A planar curve can be represented either explicitly by sampling points on the curve or using an implicit representation such as level sets (Kimmel (2012)). We work with an explicit representation of simple curves (open or closed) with random variable sampling of the points along the curve. Thus, every curve is a $N \times 2$ array denoting the $X$ and $Y$ coordinates of the $N$ points. We build a convolutional neural network which inputs a curve and outputs a representation or signature for every point on the curve. We can interpret this architecture as an algorithmic scheme of representing a function over the curve. However feeding in a single curve is insufficient and instead we run this convolutional architecture in a Siamese configuration (Figure 2) that accepts a curve and a

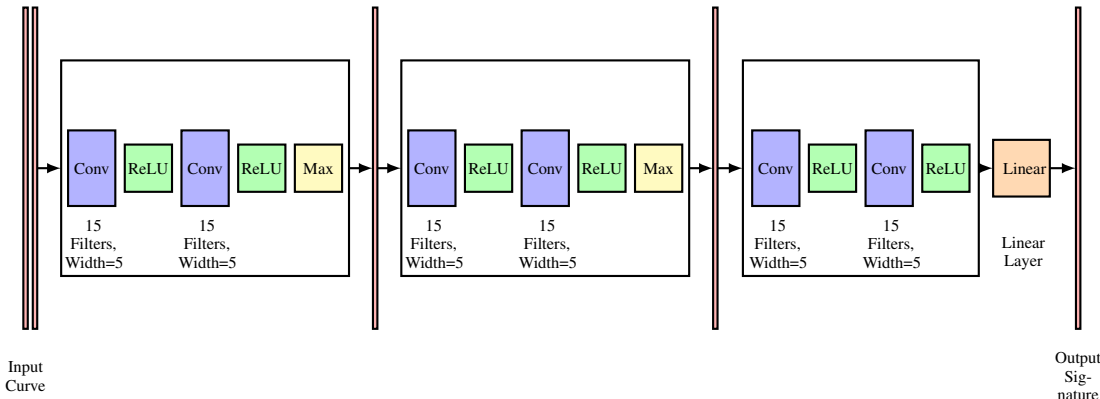

Figure 3: Network Architecture

transformed version (positive) of the curve or an unrelated curve (negative). By using two identical copies of the same network sharing weights to process these two curves we are able to extract geometric invariance by using a loss function to require that the two arms of the Siamese configuration must produce values that are minimally different for curves which are related by Euclidean transformations representing positive examples and maximally different for carefully constructed negative examples. To fully enable training of our network we build a large dataset comprising of positive and negative examples of the relevant transformations from a database of curves. We choose to minimize the contrastive loss between the two outputs of the Siamese network as this directs the network architecture to model a function over the curve which is invariant to the transformation.

## 3.1 LOSS FUNCTION

We employ the contrastive loss function (Chopra et al. (2005); LeCun et al. (2006)) for training our network. The Siamese configuration comprises of two identical networks of Figure 3 computing signatures for two separate inputs of data. Associated to each input pair is a label which indicates whether or not that pair is a positive ($\lambda = 1$) or a negative ($\lambda = 0$) example (Figure 2). Let $C_{1i}$ and $C_{2i}$ be the curves imputed to first and second arm of the configuration for the $i^{th}$ example of the data with label $\lambda_i$. Let $\mathcal{S}_{\Theta}(C)$ denote the output of the network for a given set of weights $\Theta$ for input curve $C$. The contrastive loss function is given by:

$$\mathcal{C}(\Theta) = \frac{1}{N}\Big\{ \sum_{i=1}^{i=N} \lambda_i \ ||\ \mathcal{S}_{\Theta}(C_{1i}) - \mathcal{S}_{\Theta}(C_{2i})\ ||\ + (1-\lambda_i) \ \max(\ 0,\ \mu - ||\ \mathcal{S}_{\Theta}(C_{1i}) - \mathcal{S}_{\Theta}(C_{2i})\ ||\ )\Big\}, \tag{3}$$

where $\mu$ is a cross validated hyper-parameter known as *margin* which defines the metric threshold beyond which negative examples are penalized.

## 3.2 ARCHITECTURE

The network inputs a $N \times 2$ array representing the coordinates of $N$ points along the curve. The sequential nature of the curves and the mostly $1D$-convolution operations can also be looked at from the point of view of temporal signals using recurrent neural network architectures. Here however we choose instead to use a multistage CNN pipeline. The network, given by one arm of the Siamese configuration, comprises of three stages that use layer units which are typically considered the basic building blocks of modern CNN architectures. Each stage contains two sequential batches of convolutions appended with rectified linear units (ReLU) and ending with a max unit. The convolutional unit comprises of convolutions with 15 filters of width 5 as depicted in Figure 3. The max unit computes the maximum of 15 responses per point to yield an intermediate output after each stage. The final stage is followed by a linear layer which linearly combines the responses to yield the final output. Since, every iteration of convolution results in a reduction of the sequence length, sufficient padding is provided on both ends of the curve. This ensures that the value of the signature at a point is the result of the response of the computation resulting from the filter centered around that point.

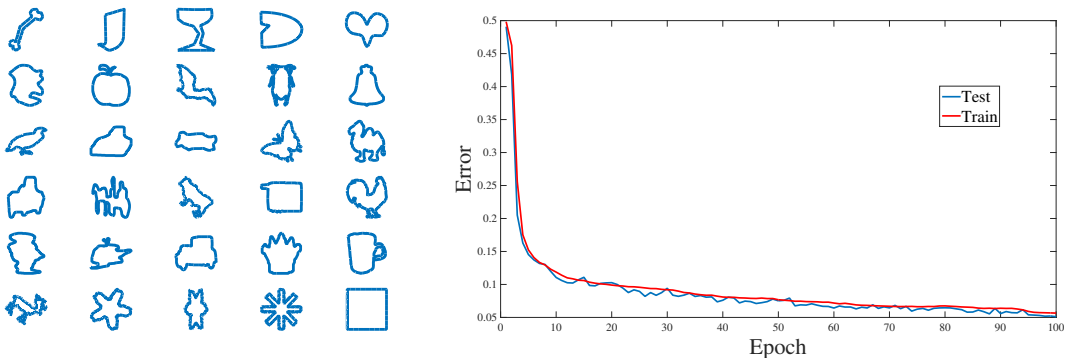

Figure 4: Contours extracted from the MPEG7 Database and the error plot for training.

### 3.3 BUILDING REPRESENTATIVE DATASETS AND IMPLEMENTATION

In order to train for invariance, we need to build a dataset with two major attributes: First, it needs to contain a large number of examples of the transformation and second, the curves involved in the training need to have sufficient richness in terms of different patterns of sharp edges, corners, smoothness, noise and sampling factors to ensure sufficient generalizability of the model. To sufficiently span the space of Euclidean transformations, we generate random two dimensional rotations by uniformly sampling angles from $[-\pi, \pi]$. The curves are normalized by removing the mean and dividing by the standard deviation thereby achieving invariance to translations and uniform scaling. The contours are extracted from the shapes of the MPEG7 Database (Latecki et al. (2000)) as shown in first part of Figure 4. It comprises a total of $1400$ shapes containing $70$ different categories of objects. $700$ of the total were used for training and $350$ each for testing and validation. The positive examples are constructed by taking a curve and randomly transforming it by a rotation, translation and reflection and pairing them together. The negative examples are obtained by pairing curves which are deemed dissimilar as explained in Section 4. The contours are extracted and each contour is sub-sampled to $500$ points. We build the training dataset of $10,000$ examples with approximately $50\%$ each for the positive and negative examples. The network and training is performed using the Torch library Collobert et al. (2002). We trained using Adagrad Duchi et al. (2011) at a learning rate of $5 \times 10^{-4}$ and a batch size of $10$. We set the contrastive loss hyperparameter *margin* $\mu = 1$ and Figure 4 shows the error plot for training and the convergence of the loss to a minimum. The rest of this work describes how we can observe and extend the efficacy of the trained network on new data.

## 4 MULTI-SCALE REPRESENTATIONS

Invariant representations at varying levels of abstraction have a theoretical interest as well as practical importance to them. Enumeration at different scales enables a hierarchical method of analysis which is useful when there is noise and hence stability is desired in the invariant. As mentioned in Section 2, the invariants constructed from scale-space methods and integral invariants, naturally allow for such a decomposition by construction.

A valuable insight for multi-scale representations is provided in the theorems of Gage, Hamilton and Grayson (Gage & Hamilton (1986); Grayson (1987)). It says that if we evolve any smooth non-intersecting planar curve with mean curvature flow, which is invariant to Euclidean transformations, it will ultimately converge into a circle before vanishing into a point. The curvature corresponding to this evolution follows a profile as shown in Figure 5, going from a possibly noisy descriptive feature to a constant function. In our framework, we observe an analogous behavior in a data-dependent setting. The positive part of the loss function ($\lambda = 1$) forces the network to push the outputs of the positive examples closer, whereas the negative part ($\lambda = 0$) forces the weights of network to push the outputs of the negative examples apart, beyond the distance barrier of $\mu$. If the training data does not contain any negative example, it is easy to see that the weights of the network will converge to a point which will yield a constant output that trivially minimizes the loss function in Equation 3.

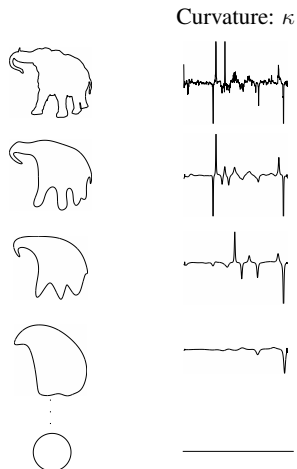

Curvature: $\kappa$

| Positive Example | Negative Example | Scale Index |
|---|---|---|
| | | Low |
| | | |
| | | |
| | | High |

Figure 5: Curve evolution and the corresponding curvature profile.

Table 1: Examples of training pairs for different scales. Each row indicates the pattern of training examples for a different scale.

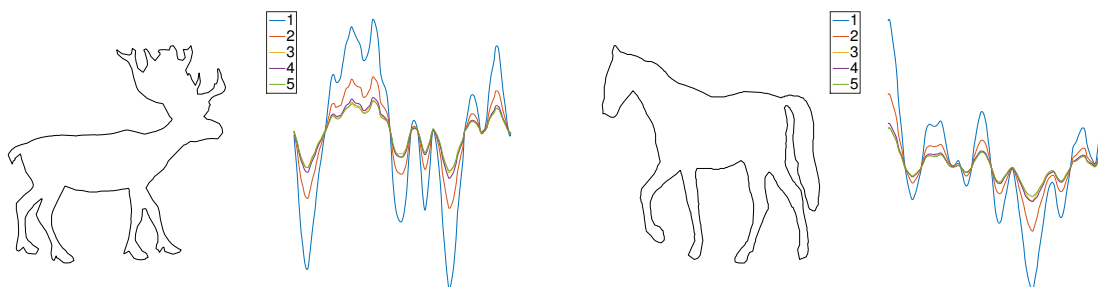

Figure 6: Experiments with multi-scale representations. Each signature is the output of a network trained on a dataset with training examples formed as per the rows of Table 1. Index1 indicates low and 5 indicates a higher level of abstraction.

This is analogous to that point in curvature flow which yields a circle and therefore has a constant curvature.

Designing the negative examples of the training data provides the means to obtain a multi-scale representation. Since we are training for a *local* descriptor of a curve, that is, a function whose value at a point depends only on its local neighborhood, a negative example must pair curves such that corresponding points on each curve must have *different local* neighborhoods. One such possibility is to construct negative examples which pair curves with their smoothed or evolved versions as in Table 1. Minimizing the loss function in equation 3 would lead to an action which pushes apart the signatures of the curve and its evolved or smoothed counterpart, thereby injecting the signature with fidelity and descriptiveness. We construct separate data-sets where the negative examples are drawn as shown in the rows of Table1 and train a network model for each of them using the loss function 3. In our experiments we perform smoothing by using a local polynomial regression with weighted linear least squares for obtaining the evolved contour. Figure 6 shows the outputs of these different networks which demonstrate a scale-space like behavior.

## 5 EXPERIMENTS AND DISCUSSION

Ability to handle low signal to noise ratios and efficiency of computation are typical qualities desired in a geometric invariant. To test the numerical stability and robustness of the invariant signatures

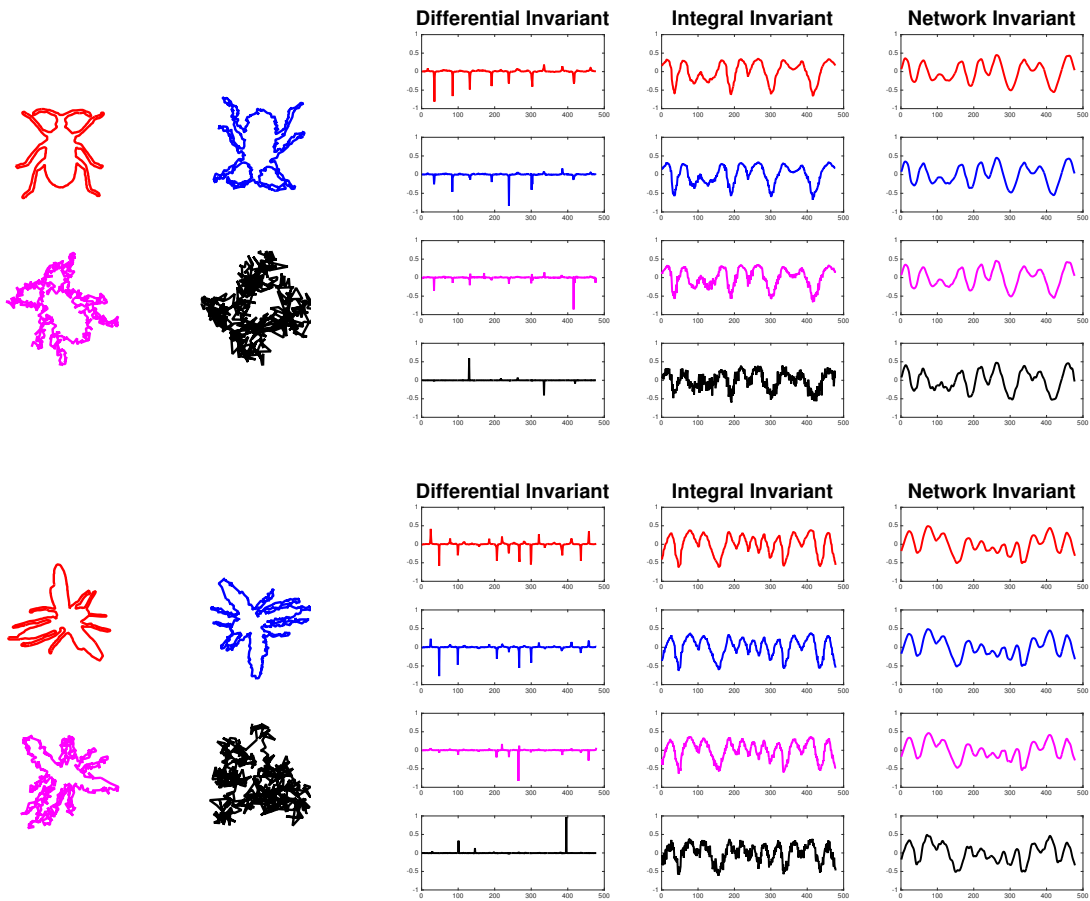

Figure 7: Stability of different signatures in varying levels noise and Euclidean transformations. The correspondence for the shape and the signature is the color. All signatures are normalized.

we designed two experiments. In the first experiment, we add increasing levels of zero-mean Gaussian noise to the curve and compare the three types of signatures: differential (Euclidean curvature), integral (integral area invariant) and the output of our network (henceforth termed as network invariant) as shown in Figure 7. Apart from adding noise, we also rotate the curve to obtain a better assessment of the Euclidean invariance property. In Figure 8, we test descriptiveness of the signature under noisy conditions in a shape retrieval task for a set of 30 shapes with 6 different categories. For every curve, we generate 5 signatures at different scales for the integral and the network invariant and use them as a representation for that shape. We use the Hausdorff distance as a distance measure (Bronstein et al. (2008)) between the two sets of signatures to rank the shapes for retrieval. Figure 7 and 8 demonstrate the robustness of the network especially at high noise levels.

In the second experiment, we decimate a high resolution contour at successive resolutions by randomly sub-sampling and redistributing a set of its points (marked blue in Figure 9) and observe the signatures at certain *fixed* points (marked red in Figure 9) on the curve. Figure 9 shows that the network is able to handle these changes in sampling and compares well with the integral invariant. Figures 7 and Figure 9 represent behavior of geometric signatures for two different tests: large noise for a moderate strength of signal and low signal for a moderate level of noise.

## 6 CONCLUSION

We have demonstrated a method to learn geometric invariants of planar curves. Using just positive and negative examples of Euclidean transformations, we showed that a convolutional neural network

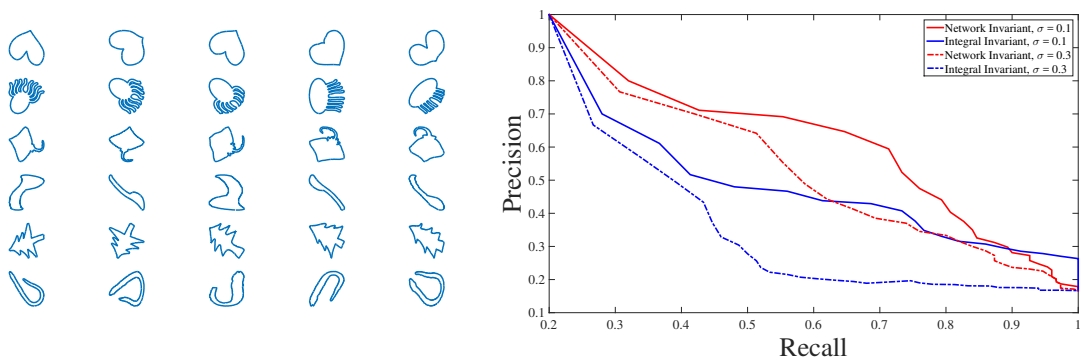

Figure 8: 5 shape contours of 6 different categories and the shape retrieval results for this set for different noise levels.

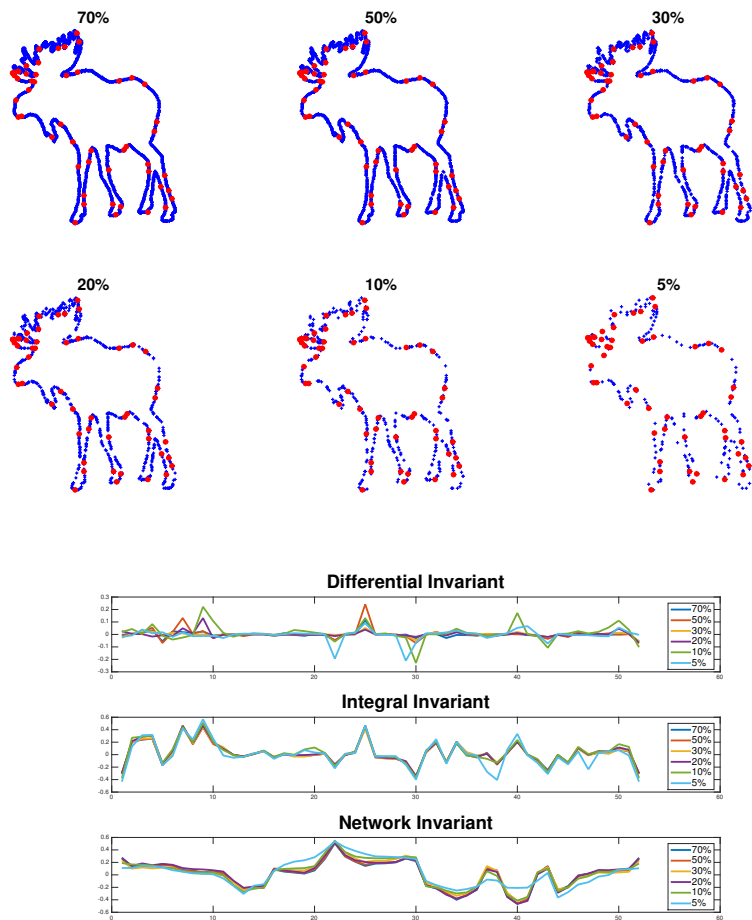

Figure 9: Testing robustness of signatures to different sampling conditions. The signatures are evaluated at the *fixed red* points on each contour and the density and distribution of the blue points along the curve is varied from 70% to 5% of the total number of points of a high resolution curve.

is able to effectively discover and encode transform-invariant properties of curves while remaining numerically robust in the face of noise. By using a geometric context to the training process we were able to develop novel multi-scale representations from a learning based approach without explicitly

enforcing such behavior. As compared to a more axiomatic framework of modeling with differential geometry and engineering with numerical analysis, we demonstrated a way of replacing this pipeline with a deep learning framework which combines both these aspects. The non-specific nature of this framework can be seen as providing the groundwork for future deep learning data based problems in differential geometry.

ACKNOWLEDGMENTS

This project has received funding from the European Research Council (ERC) under the European Unions Horizon 2020 research and innovation program (grant agreement No 664800)

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

## 7    APPENDIX

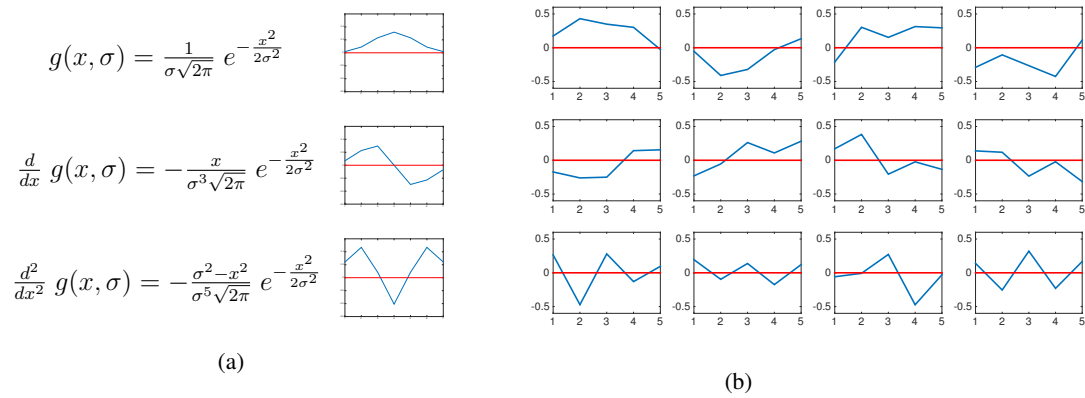

$$g(x,\sigma) = \frac{1}{\sigma\sqrt{2\pi}}\, e^{-\frac{x^2}{2\sigma^2}}$$

$$\frac{d}{dx}\, g(x,\sigma) = -\frac{x}{\sigma^3\sqrt{2\pi}}\, e^{-\frac{x^2}{2\sigma^2}}$$

$$\frac{d^2}{dx^2}\, g(x,\sigma) = -\frac{\sigma^2-x^2}{\sigma^5\sqrt{2\pi}}\, e^{-\frac{x^2}{2\sigma^2}}$$

(a)

(b)

Figure 10: (a) Standard 1D Gaussian filters and its derivatives used for curvature and curvature scale space calculations. (b) Some of the filters from the first layer of the network proposed in this paper. One can interpret the shapes of the filters in (b) as derivative kernels which are learned from data and therefore adapted to its sampling conditions.

