# Peer review of "Learning Invariant Representations Of Planar Curves "

_ICLR 2017 — accepted_

[Official Review · AnonReviewer3 · rating 6 · confidence 5 · 16 Dec 2016]
**filling a much needed gap?**

I'm torn on this one. Seeing the MPEG-7 dataset and references to curvature scale space brought to mind the old saying that "if it's not worth doing, it's not worth doing well." There is no question that the MPEG-7 dataset/benchmark got saturated long ago, and it's quite surprising to see it in a submission to a modern ML conference. I brought up the question of "why use this representation" with the authors and they said their "main purpose was to connect the theory of differential geometry of curves with the computational engine of a convolutional neural network." Fair enough. I agree these are seemingly different fields, and the authors deserve some credit for connecting them. If we give them the benefit of the doubt that this was worth doing, then the approach they pursue using a Siamese configuration makes sense, and their adaptation of deep convnet frameworks to 1D signals is reasonable. To the extent that the old invariant based methods made use of smoothed/filtered representations coupled with nonlinearities, it's sensible to revisit this problem using convnets. I wouldn't mind seeing this paper accepted, since it's different from the mainstream, but I worry about there being too narrow an audience at ICLR that still cares about this type of shape representation.

[Official Review · AnonReviewer2 · rating 5 · confidence 2 · 16 Dec 2016 (modified: 17 Jan 2017)]
**Limited theoretical novelty and evaluation**

Authors show that a contrastive loss for a Siamese architecture can be used for learning representations for planar curves. With the proposed framework, authors are able to learn a representation which is comparable to traditional differential or integral invariants, as evaluated on few toy examples.

The paper is generally well written and shows an interesting application of the Siamese architecture. However, the experimental evaluation and the results show that these are rather preliminary results as not many of the choices are validated. My biggest concern is in the choice of the negative samples, as the network basically learns only to distinguish between shapes at different scales, instead of recognizing different shapes. It is well known fact that in order to achieve a good performance with the contrastive loss, one has to be careful about the hard negative sampling, as using too easy negatives may lead to inferior results. Thus, this may be the underlying reason for such choice of the negatives? Unfortunately, this is not discussed in the paper.

Furthermore the paper misses a more thorough quantitative evaluation and concentrates more on showing particular examples, instead of measuring more robust statistics over multiple curves (invariance to noise and sampling artifacts).

In general, the paper shows interesting first steps in this direction, however it is not clear whether the experimental section is strong and thorough enough for the ICLR conference. Also the novelty of the proposed idea is limited as Siamese networks are used for many years and this work only shows that they can be applied to a different task.

[Official Review · AnonReviewer1 · rating 8 · confidence 3 · 16 Dec 2016]
**An interesting representation**

Pros : 
- New representation with nice properties that are derived and compared with a mathematical baseline and background
- A simple algorithm to obtain the representation

Cons :
- The paper sounds like an applied maths paper, but further analysis on the nature of the representation could be done, for instance, by understanding the nature of each layer, or at least, the first.

[Author Response · Gautam Pai · 27 Dec 2016]
**Changes in the updated draft**

Based on the constructive suggestions of the reviewers, we have updated the paper with two minor changes:

(1.) We have added a figure in the appendix section, showing the learned filters from the first layer of the network.

(2.) We have added an additional line in Section 4:  "Since we are training for a local descriptor of a curve, that is, a function whose value at a point depends only on its local neighborhood, a negative example must pair curves such that corresponding points on each curve must have different local neighborhoods",  in order to highlight the locality property of our framework.

[Final Decision · Program Chairs · 06 Feb 2017]
**ICLR committee final decision**

This work proposes learning of local representations of planar curves using convolutional neural networks.
 Invariance to rigid transformations and discriminability are enforced with a metric learning framework using a siamese architecture. Preliminary experiments on toy datasets compare favorably with predefined geometric invariants (both differential and integral). 
 
 The reviewers found value on the problem set-up and the proposed model, and were generally satisfied with the author's response. They also expressed concern that the experimental section is currently a bit weak and does not include any real data. Also, the paper does not offer theoretical insights that inform us about the design of the representation or about the provable invariance guarantees. All things considered, the AC recommends acceptance in the form of a poster, but strongly encourages the authors to strengthen the work both in the experimental and the theoretical aspects.